# Loss-of-Function Variants in *DRD1* in Infantile Parkinsonism-Dystonia

**DOI:** 10.3390/cells12071046

**Published:** 2023-03-30

**Authors:** Kimberley M. Reid, Dora Steel, Sanjana Nair, Sanjay Bhate, Lorenzo Biassoni, Sniya Sudhakar, Michelle Heys, Elizabeth Burke, Erik-Jan Kamsteeg, Biju Hameed, Michael Zech, Niccolo E. Mencacci, Katy Barwick, Maya Topf, Manju A. Kurian

**Affiliations:** 1Molecular Neurosciences, Developmental Neurosciences, Zayed Centre for Research into Rare Disease in Children, UCL GOS Institute of Child Health, London WC1N 1DZ, UK; kimberley.reid@ucl.ac.uk (K.M.R.); d.steel@ucl.ac.uk (D.S.); k.barwick@ucl.ac.uk (K.B.); 2Department of Neurology, Great Ormond Street Hospital, London WC1N 3JH, UK; sanjay.bhate@gosh.nhs.uk (S.B.); l.biassoni@gosh.nhs.uk (L.B.); sniya.sudhakar@gosh.nhs.uk (S.S.); biju.hameed@gosh.nhs.uk (B.H.); 3Leibniz Institute of Virology (LIV), Centre for Structural Systems Biology (CSSB), 20251 Hamburg, Germany; sanjana.nair@cssb-hamburg.de (S.N.);; 4Department of Radiology, Great Ormond Street Hospital, London WC1N 3JH, UK; 5Department of Population, Policy and Practice, UCL GOS Institute of Child Health, London WC1N 1DZ, UK; m.heys@ucl.ac.uk; 6Specialist Children’s and Young People’s Services, Newham, East London NHS Foundation Trust, London RM13 8GQ, UK; 7Office of the Clinical Director, National Human Genome Research Institute, Bethesda, MD 20892, USA; elizabeth.burke2@nih.gov; 8Undiagnosed Diseases Program and Network, Office of the Director, National Institutes of Health, Bethesda, MD 20892, USA; 9Department of Human Genetics, Radboud University Medical Center, 6525 Nijmegen, The Netherlands; erik-jan.kamsteeg@radboudumc.nl; 10Genomics England, London EC1M 6BQ, UK; gecip@genomicsengland.co.uk; 11William Harvey Research Institute, Queen Mary University of London, London EC1M 6BQ, UK; 12Institute of Human Genetics, School of Medicine, Technical University of Munich, 85354 Munich, Germany; michael.zech@mri.tum.de; 13Institute of Neurogenomics, Helmholtz Zentrum München, 85764 Munich, Germany; 14Feinberg School of Medicine, Northwestern University, Chicago IL 60611, USA; niccolo.mencacci@northwestern.edu

**Keywords:** DRD1, dopamine, dystonia, parkinsonism

## Abstract

The human dopaminergic system is vital for a broad range of neurological processes, including the control of voluntary movement. Here we report a proband presenting with clinical features of dopamine deficiency: severe infantile parkinsonism-dystonia, characterised by frequent oculogyric crises, dysautonomia and global neurodevelopmental impairment. CSF neurotransmitter analysis was unexpectedly normal. Triome whole-genome sequencing revealed a homozygous variant (c.110C>A, (p.T37K)) in *DRD1*, encoding the most abundant dopamine receptor (D_1_) in the central nervous system, most highly expressed in the striatum. This variant was absent from gnomAD, with a CADD score of 27.5. Using an in vitro heterologous expression system, we determined that *DRD1*-T37K results in loss of protein function. Structure-function modelling studies predicted reduced substrate binding, which was confirmed in vitro. Exposure of mutant protein to the selective D_1_ agonist Chloro APB resulted in significantly reduced cyclic AMP levels. Numerous D_1_ agonists failed to rescue the cellular defect, reflected clinically in the patient, who had no benefit from dopaminergic therapy. Our study identifies *DRD1* as a new disease-associated gene, suggesting a crucial role for the D_1_ receptor in motor control.

## 1. Introduction

Dopamine (DA) is a key monoamine neurotransmitter involved in initiation and control of voluntary movement, motor learning, motivation and executive function [1]. Within the CNS, DA is primarily synthesised in the midbrain (substantia nigra, ventral tegmental area), with extensive projections to the striatum, cortex, nucleus accumbens and putamen [1]. In the dorsal striatum, DA acts on cell-surface dopamine receptors of medium spiny neurons (MSNs)—GABAergic inhibitory neurons which form the major efferent projections of the basal ganglia—in the direct pathway (reducing inhibitory output of the globus pallidus interna and substantia nigra to the thalamus) and indirect pathway (increasing inhibitory output) (Figure 1) [2]. There are five dopamine receptor subtypes (D_1_–D_5_), divided into two classes: D_1_-like (D_1_ and D_5_) and D_2_-like (D_2_, D_3_ and D_4_). All are G-protein-coupled receptors (GPCRs) but D_1_-like receptors are coupled with G_s𝛼_ proteins, where DA binding stimulates adenylyl cyclase to increase cyclic adenosine monophosphate (cAMP) levels, leading to, among other things, activation of protein kinase A (PKA) and generally an excitatory response, whereas D_2_-like receptors are coupled with G_i_ proteins, which inhibit adenylyl cyclase, reducing cAMP and exerting an inhibitory effect (Figure 2) [3]. The direct (prokinetic) pathway consists primarily of D_1_-expressing MSNs whilst the indirect (antikinetic) pathway has mainly D_2_-expressing MSNs [4,5]. Broadly speaking, D_1_ activation increases motor activity whereas D_2_ activation reduces it.

Defective DA homeostasis is linked to a broad range of neurological and neuropsychiatric diseases. A number of rare monogenic neurotransmitter diseases are associated with impaired DA synthesis, metabolism, transport and signalling (Figure 2); these disorders are typically characterised by neurodevelopmental delay, complex hyper- and hypokinetic movement phenotypes and mood/behaviour symptoms [6]. DA dysregulation is implicated in a broad range of common diseases from Parkinson’s disease (PD) and restless leg syndrome to neuropsychiatric disorders such as depression, schizophrenia, and attention deficit hyperactivity disorder (ADHD). Polymorphisms in DA receptor genes have been implicated in addiction, psychiatric illness and PD [7,8] and more recently heterozygous gain-of-function variants in the D_2_ receptor have been reported in patients with a hyperkinetic movement disorder [9,10]. In this study, we describe a new dopamine receptor disorder associated with recessive loss-of-function variants in the gene encoding the D_1_ receptor, *DRD1*.

## 2. Materials and Methods

### 2.1. Ethics and Consent

The proband’s parents provided written informed consent for whole-genome sequencing (WGS) on a research basis (Bloomsbury Local Research Ethics Committee 13LO168). Separate written consent was given for publication of video footage.

### 2.2. Clinical Characterisation

A detailed history was taken from the parents and the proband was clinically examined by members of the clinical team. Additional information was obtained from the electronic patient records.

### 2.3. Molecular Genetic Analysis

Lymphocytic DNA was extracted from peripheral blood from the proband and both parents. WGS was performed by BGI Group, using the DNBSeq^TM^ NGS technology platform, with paired-end 100 bp reads to a target depth of 30×. Variant calling from Fastq files was performed by Novogene: reads were aligned to the GRCh38 reference genome and variant calling was undertaken using SAMtools [11], SOAPsnp [12] and GATK [13].

Variant prioritisation and filtering were undertaken using QIAGEN Ingenuity Variant Analysis^TM^. Homozygous and compound heterozygous variants with a minor allele frequency (MAF) in gnomAD (https://gnomad.broadinstitute.org/, accessed on 1 December 2022) of <1% and de novo heterozygous variants with MAF < 0.01%, which were exonic or within 20 bps of an exon, were longlisted. Initial analysis used an extended panel of 3746 genes associated with known neurological disease phenotypes (Appendix A). After this, panel-free genomic analysis was undertaken. Candidate disease-associated variants were selected using AlamutVisual^®^ 2.11, with prioritisation based on predicted impact on protein function using in silico tools ((CADD score [14], SIFT (v.6.2.0), MutationTaster (v.2021) and PolyPhen-2 (v.2.2.r398)) and on potential relevance of the gene product to a neurodevelopmental disorder (i.e., central nervous system expression, physiological function and any known impact of disruption in humans, animal or cellular models). The *DRD1* variant was confirmed in the proband and parents using Sanger sequencing, as previously published [15], using forward and reverse primers, DRD1_F 5′-cggtcctctcatggaatgtt-3′ and DRD1_R 5′-gtgtcagatctcttggtggc-3′.

Following identification of the candidate gene, seven international genomic databases with over 120,000 exomes or genomes were interrogated for biallelic variants in *DRD1*. These included three large databases of individuals with a range of rare diseases (Deciphering Developmental Disorders study [16]; NIH Undiagnosed Diseases Program [17]; Genomics England 100,000 Genomes Project [18]) and three databases of patients with undiagnosed neurological disorders (Radboud University Medical Centre Nijmegen [19]; the Institut für Humangenetik, Munich [20] and the Istituto Neurologico “Carlo Besta”, Milan [10]).

### 2.4. Site-Directed Mutagenesis, Cell Culture and Transfection

The pCMV6 plasmid with FLAG- and Myc-tagged human dopamine receptor D1 (*DRD1*) (Origene, cat# RC210389) was utilised for site-directed mutagenesis using the QuikChange Lightning site-directed mutagenesis kit (Stratagene, Santa Clara, CA, USA) with primers *DRD1*_T37K_F 5′-gttccccaggagcttggacaggatgagca-3′ and *DRD1*_T37K_R 5′-tgctcatcctgtccaagctcctggggaac-3′. Clones containing the desired variant were sequence verified and expanded using the Qiagen Maxi-prep kit according to manufacturer instructions. The presence of the intended sequence was confirmed by Sanger sequencing.

pCMV6-*DRD1*-WT or pCMV6-*DRD1*-T37K was transiently transfected into HEK-293T cells using Lipofectamine 2000 according to manufacturer instructions, using a 1:2 ratio of DNA to lipofectamine. Such heterologous in vitro overexpression cell-model systems are commonly used to determine the effect of disease-associated gene variants on gene function [21,22,23,24]. Cells were cultured on Poly-D-Lysine coated or uncoated tissue culture flasks in Dulbecco’s Modified Eagle Medium (DMEM) supplemented with 10% fetal bovine serum (FBS), 100 mg/L penicillin/streptomycin and 2 mM L-glutamine at 37 °C supplemented with 5% CO_2_ for 24–48 h before analysis.

### 2.5. Western Blotting

Untransfected HEK-293T cells, cells transiently expressing either pCMV6-*DRD1*-WT or pCMV6-*DRD1*-T37K, and cells transfected with the pCMV6 empty vector, were washed with cold PBS and lysed in RIPA buffer (ThemoFisher, Massachusetts) plus protease inhibitor cocktail for 30 min at 4 °C with inversion. Lysates were centrifuged at 140,000× *g* for 10 min at 4 °C to remove cell debris. Lysates were then resolved on 4–12% Bis-Tris NuPAGE gels at 150 V for 60 min, before being transferred to PVDF membranes. D_1_ was probed with a mouse monoclonal Myc-tag antibody (CST cat# 2276), followed by HRP-conjugated goat anti-mouse secondary antibody. A polyclonal β-actin antibody (Sigma-Aldrich, St. Louis, MO, USA) was used as a loading control. Signal was detected using the SuperSignal West Dura Extended Duration chemiluminescence substrate solution (Thermo Scientific, Waltham, MA, USA) according to manufacturer instructions and blots were imaged on a BioRad ChemiDoc MP imaging system. The relative expression of *DRD1*-WT or *DRD1*-T37K was normalised to β-actin, and comparisons performed using an unpaired parametric Student’s *t*-test. Statistical significance was considered where *p* < 0.05.

### 2.6. D_1_ Surface Expression by Biotinylation

Untransfected HEK-293T cells or those transiently expressing either pCMV6 empty vector, pCMV6-*DRD1*-WT or pCMV6-*DRD1*-T37K were washed with cold CM-PBS (100 mg/L MgCl_2_.6H_2_O and 100 mg/L CaCl_2_ in 1X PBS) before incubation with Sulfo-NHS-SS-biotin (0.5 mg/mL in CM-PBS; Pierce Biotechnology) for 30 min on ice. Biotin was then quenched with quenching buffer (50 mM NH_4_Cl in CM-PBS). Cells were lysed in RIPA buffer plus protease inhibitor cocktail for 30 min at 4 °C with inversion. The lysate samples were centrifuged at 14,000× *g* for 10 min at 4 °C to remove cell debris. This formed the input fraction. The biotinylated proteins were then separated with immobilised monomeric NeutrAvidin beads (Pierce Biotechnology) and eluted with SDS-PAGE sample buffer and boiling. The total lysates and biotinylated proteins were resolved on 4–12% Bis-Tris NuPAGE gels. D_1_ was probed with a mouse monoclonal Myc-tag antibody, followed by HRP-conjugated goat anti-mouse antibody. Polyclonal β-actin antibody (Sigma-Aldrich, St. Louis, MO, USA) and Na^+^/K^+^ α ATPase antibody (Abcam) were used as loading controls for the total cell lysate and membrane-bound fractions, respectively. The transporter signal was visualised using SuperSignal West Dura Extended Duration chemiluminescence substrate solution (Thermo Scientific). The relative expression of the cell surface D_1_ fraction was normalised to Na^+^/K^+^ α ATPase expression. Comparisons were performed using an unpaired parametric Student’s *t*-test. Statistical significance was considered where *p* < 0.05.

### 2.7. D_1_ Surface Expression by Immunofluorescence

Untransfected HEK-293T cells and cells transiently expressing either pCMV6 empty vector, pCMV6-*DRD1*-WT or pCMV6-*DRD1*-T37K were cultured on a 13 mm glass coverslip for 24 h then washed with PBS and fixed for 10 min using 4% PFA. Cells were then washed again in PBS and incubated in immunofluorescence permeabilization buffer (PBS plus 0.1% Triton) for 10 min at room temperature. Cells were washed thrice in PBS and incubated with 10% FBS in PBS for 1 h to block. D_1_ was probed with mouse monoclonal Myc-tag antibody (CST cat# 2276) at 1:8000 dilution overnight at 4 °C. Cells were washed in PBS before being incubated with a goat anti-mouse H&L Alexa Fluor 594 secondary antibody for 1 h at room temperature. Cells were again washed in PBS and incubated with DAPI and wheat-germ agglutinin (WGA) in PBS for 5 min before being mounted on SuperFrost microscope slides with Prolong Gold mounting media. Cells were imaged using a Zeiss LSM710 confocal microscope.

### 2.8. cAMP Assay

Untransfected HEK-293T cells or those transiently expressing either pCMV6 empty vector, pCMV6-*DRD1*-WT or pCMV6-*DRD1*-T37K were cultured on white, clear-bottom 96-well plates for 24 h. Cells were briefly washed with PBS to remove traces of serum and then incubated with concentrations of Chloro APB, pramipexole, apomorphine, rotigotine or bromocriptine in PBS that contained 100 µM IBMX and 100 µM Ro20-1724 for 30 min at 37 °C. cAMP levels were then recorded using the cAMP Glo Max assay (Promega) according to the manufacturer’s instructions. Briefly, cells were lysed with the cAMP detection solution plus protein kinase A for 20 min at room temperature. An equal volume of Kinase-Glo was added and incubated for 10 min at room temperature, and plates were read after 10 min using a HidEx Sense luminometer. For analysis, values were inverted for clear graphical representation of cAMP production. Comparisons were conducted using a two-way ANOVA with multiple comparisons. Statistical significance was considered where *p* < 0.05.

### 2.9. Radiolabelled Ligand Binding Assay

Cell-free membrane preparations were first prepared from HEK-293T cells transiently expressing either pCMV6-*DRD1*-WT or pCMV6-*DRD1*-T37K. Twenty-four hours post transfection, cells grown in T75 culture flasks were washed in PBS before being detached using Trypsin-EDTA. After a second wash in PBS, cells were lysed using ice-cold lysis buffer (50 mM Tris-HCl, 5 mM MgCl_2_, 5 mM EDTA, protease inhibitor cocktail). Lysates were then centrifuged at 17,000× *g* for 10 min at 4 °C to pellet the membranes. The pellet was then homogenised in fresh lysis buffer and centrifuged for a second time at 17,000× *g* for 10 min at 4 °C. The pellet was then homogenised again in sucrose lysis buffer (50 mM Tris-HCl, 5 mM MgCl_2_, 5mM EDTA, protease inhibitor cocktail, 10% sucrose) before freezing at −80 °C until needed. Protein concentration of the membrane preparation was determined using a Pierce BCA assay.

In a total reaction volume of 250 µL set up in 96-well plates, 75 µg of membrane preparations were mixed with increasing concentrations of tritiated DA (ranging from 0.01 µM to 300 µM) in assay binding buffer (50 mM Tris, 5 mM MgCl_2_, 0.1 mM EDTA, pH 7.4) and incubated at 25 °C for 10 min. The incubation was terminated by filtration and subsequent washes through a Filtermat using a Tomtec cell harvester. Filtermats were sealed with Meltilex. The radioactivity of the Filtermats was then assessed using a solid scintillation MicroBeta counter. Scintillation levels were normalised to membrane preparations similarly extracted from untransfected control HEK-293T cells. Results were analysed using GraphPad Prism V9.0.0 using “One site-Total” non-linear regression. Comparisons between groups was performed using a two-way ANOVA. Statistical significance was considered where *p* < 0.05.

### 2.10. Molecular Modelling

The structure of the wildtype DRD1 was available in the Protein Data Bank (PDB) with PDB ID: 7JVP. The model of the T37K mutant protein was generated by homology modelling using the default parameters in ModWeb (http://salilab.org/modweb, accessed on 1 December 2022). The template detected by ModWeb had the PDB ID: 7JOZ, which is another experimentally derived structure of DRD1. The angles between the TM1 and TM2 helices were calculated in Chimera. An axis was created for each helix and the angle between the axes was used to derive the angle between the helices. The coulombic potential was generated using the ‘Coulombic Surface Coloring’ module of Chimera.

## 3. Results

### 3.1. Clinical Phenotyping Reveals a Complex Infantile Parkinsonism-Dystonia Phenotype

The proband is the first child of healthy consanguineous Afghani parents (Figure 3A). She presented at six months old with developmental delay and never acquired the ability to sit, roll or babble. From infancy onwards she experienced recurrent episodes of severe generalised dystonia lasting for hours, sometimes triggered by pain, anxiety or excitement. Other symptoms included constipation, excessive sweating and chronic nasal congestion. Enteral feeding was commenced at three years of age because of an unsafe swallow and she required treatment for gastro-oesophageal reflux.

Examined at four years old, she was normocephalic with no dysmorphic features. Antigravity movements were present but there was a marked paucity of spontaneous movement. She was hypomimic but appeared visually alert and demonstrated a responsive smile. Truncal tone was low, with head lag, but limb tone was variable, with fluctuating generalised dystonia, brisk tendon reflexes, downgoing plantars and striatal toe. The reported dystonic episodes were characterised by neck arching, limb posturing and upward eye deviation, suggesting oculogyric crises (Appendix A, Figure 3B–D).

There was no clinical response to levodopa at up to 10 mg/kg/day and only modest improvement with other tone-modifying agents, although transdermal clonidine yielded some improvement in her dystonia.

There was a strong clinical suspicion that she was likely to have a monoamine neurotransmitter disorder. Surprisingly, CSF monoamine neurotransmitter levels and AADC enzyme activity were normal. Extensive neurometabolic investigations, a genetic panel for childhood-onset movement disorders, EEG and imaging including MRI and DaTscan were also non-contributory.

### 3.2. Molecular Genetic Analysis Identifies DRD1 as a Candidate Gene

Initial analysis of whole-genome sequencing (WGS) data from the proband and unaffected parents, using a panel of 3476 genes known to be associated with neurological disorders, did not identify any likely pathogenic variant. A heterozygous de novo variant in *PABPC1* (NM_002568.4:c.388-1G>A, p.?) was considered but was felt unlikely to explain the proband’s phenotype because it occurred at low frequencies in some normal population databases (130 occurrences in ExAc [25], absent from gnomAD). Missense variants in *PABPC1* have been reported in developmental disorders, but these variants are clustered in a different region of the gene and most affected patients had a much milder phenotype [26], characterised by moderate developmental delay but with achievement of ambulation in most cases.

Panel-free analysis did not identify any de novo or compound heterozygous variants likely to be relevant. Nineteen homozygous variants were detected in the proband, predicted to affect the protein sequence either through altering the coding sequence or occurring near a splice site, with a read depth of ≥20× and an MAF of <1%. Of these, 12 had a CADD score < 15, indicating very low likelihood of pathogenicity. The remaining 7 variants (in *ATXN3*, *CIC*, *DRD1, GPR4, MUC16, ZAN* and *ZNF354C*) are detailed in Appendix A. The variants in *CIC* and *MUC16* could be confidently excluded as candidates for severe childhood-onset disease because both occur in homozygosity in normal population databases including gnomAD. ZAN is expressed exclusively in spermatozoa. The *ATXN3* variant reflects the variability of the CAG repeat region of this gene—frameshifts here are common and the insertion length did not approach the threshold for a pathological triplet repeat expansion as found in Machado-Joseph disease [27] (which would also not match the proband’s presentation). The remaining three variants (in *DRD1*, *GPR4* and *ZNF354C*) must be considered as possible candidates for pathogenicity but we considered *DRD1* to be the most probable. ZNF354C is a ubiquitously expressed zinc finger transcription repressor: it is involved in control of bone development endothelial sprouting but has relatively low brain expression and no known specific contribution to neurological functioning [28]. GPR4 is a proton-sensing GPCR; overall brain expression is very low although it is expressed in a specific population of medullary neurons that help regulate breathing [29]. The variant occurs close to the 3′ terminus of the gene, where truncating variants may have a reduced impact on protein function.

The *DRD1* variant, NM_000794.4:c.110C>A, p.T37K, was considered a strong candidate for two reasons. First, it appeared likely to impact protein function; it affects a highly conserved amino acid residue which exhibits species conservation to zebrafish (Figure 3E), and all in silico tools used predicted a damaging effect (CADD: 27.5, PolyPhen 2 (HumVar): 0.99 (probably damaging), SIFT: 0.02 (deleterious), Mutation Taster: deleterious (89|11) (Appendix A). It is also absent from normal population databases such as gnomAD. Second, the physiological function of the gene product, D_1_, was highly relevant to the proband’s presenting symptoms, which before genetic analysis was undertaken had already been thought to reflect a deficiency of dopaminergic function.

Sanger sequencing of the DRD1-T37K variant confirmed that it was homozygous in the proband and heterozygous in both parents (Figure 3F). No additional individuals with biallelic pathogenic *DRD1* variants were identified through interrogation of additional genome and exome databases.

### 3.3. DRD1-T37K Showed No Statistical Differences in Total Protein Expression and Cell Surface Localisation

In HEK-293T cells transfected with either DRD1-WT or DRD1-T37K, total expression of D_1_ protein showed reduced levels, but not reaching statistical significance (Appendix A). Biotinylation studies showed comparable levels of cell-surface protein with no statistically significant difference in cell surface localisation (Appendix A). This was also confirmed by immunohistochemistry (Appendix A).

### 3.4. DRD1-T37K Affected the cAMP Response on D_1_ Activation

Once D_1_ is activated by binding of an extracellular agonist, adenylyl cyclase is activated, converting ATP to cAMP. To assess the function of DRD1-T37K, untransfected cells and cells transfected with DRD1-WT, DRD1-T37K or an empty vector were treated with increasing concentrations of the synthetic D_1_-specific agonist Chloro-APB, and cAMP levels recorded using the cAMP GloMax assay. Untransfected cells and cells transfected with an empty vector showed no cAMP increase in response to Chloro APB. Cells expressing DRD1-WT showed an increase in cAMP with Chloro APB treatment, which was further elevated with increasing Chloro APB concentrations. Cells expressing DRD1-T37K showed significantly less cAMP production upon Chloro APB treatment in comparison to the WT (n = 6) (Figure 4A), indicating loss of receptor function.

### 3.5. Homology Modelling of DRD1-T37K Predicted Altered Ligand Binding, Confirmed In Vitro

D_1_ belongs to the G_sα_ family of GPCRs, which all have similar structures; an N-terminal cytoplasmic domain, seven membrane spanning a helices, and an intracellular C-terminus [30,31,32]. Thr37 is located on transmembrane domain (TM) 1. The mutation of this threonine to lysine changes the relative orientation of TM1 and TM2 due to the comparatively larger size of lysine (Figure 4B,C). While the wildtype protein has an angle of 32.3° between the TM1 and TM2 helices, this is reduced to 23° in the T37K mutant. Thr also forms a hydrophobic pocket with Ser325 and Ser324 on TM7, which interacts directly with the ligand binding site (Figure 4D,E). Introduction of the positive charge through lysine changes the properties of the pocket (Figure 4F,G). The conformational change in TM1–TM2 and the addition of positive charge in the hydrophobic pocket is predicted to affect ligand binding.

To validate the findings on homology modelling, we next determined the effect of DRD1-T37K on ligand binding using a tritiated (^3^H) DA radiolabelled ligand binding assay. Cell-free membrane preparations were generated from cells transiently expressing either WT or DRD1-T37K, before incubation with increasing concentrations of (^3^H) DA for 10 min. A significant reduction in ligand binding was observed in cells expressing DRD1-T37K compared to wildtype (Figure 4H), suggesting that the mutant negatively impacts ligand binding.

### 3.6. DRD1-T37K Function Cannot Be Rescued by Dopamine Agonists

In order to determine whether DRD1-T37K function could be rescued, we investigated the effects of different dopaminergic agonists on the cAMP response. Cells expressing DRD1-WT showed the expected dose-related cAMP increase in response to the agonists rotigotine, apomorphine, bromocriptine and pramipexole but cells expressing DRD1-T37K showed no significant difference to untransfected cells (Figure 4I–L).

## 4. Discussion

We report a patient with classical infantile parkinsonism-dystonia in whom we identified a homozygous variant in *DRD1* resulting in loss-of-function of the D_1_ receptor, as demonstrated through an in vitro overexpression HEK-293T cell system. To our knowledge, this is the first report of monogenic human disease associated with D_1_ receptor dysfunction. Despite our efforts, only one affected individual has so far been found and therefore the gene-disease relationship cannot be considered fully confirmed. However, we consider *DRD1* to be a highly plausible candidate for pathogenicity. The D_1_ receptor is an essential element of the direct pathway, by which the basal ganglia promote motor activity. Loss-of-function in this pathway would be expected to result in a paucity of spontaneous movement, just as seen in our profoundly parkinsonian proband. The hyperkinetic elements of her condition, such as oculogyric and dystonic crises, are clinically indistinguishable from those seen in known monoamine disorders involving loss of dopaminergic function such as AADC deficiency.

This complex motor phenotype is reflected in animal models of loss of D_1_ function. Abnormalities of motor function and/or learning are features of multiple different mouse knockout (KO) models, but their nature varies. In *Drd1* KO mice generated using homologous recombination techniques to replace 95% of the gene with a neomycin cassette, Xu et al. reported reduced growth and increased locomotor activity (photobeam interruptions more than doubled compared with wildtype controls) [33]. In contrast, two other studies of KO mice created independently but using a similar technique reported severe feeding difficulties in infancy, lethal soon after weaning unless special softened feed was provided, with subtle deficits in motor activity in adulthood (reduced rearing activity with an impression of overall reduced activity [26], or reduced movement initiation, together with inability to improve performance over time in a Morris water maze) [27]. Another KO, generated in 2005 by similar methods, showed severely reduced and slowed locomotor activity combined with slower improvement in a place-learning task [34]. However, in 2014, using the same model, Nakamura et al. [35] reported that the *Drd1* KO showed increased spontaneous locomotor activity compared with wildtype, although it had significant impairments in rotarod and step-wheel tasks. Using an alternative approach, Durieux et al. [36] selectively ablated D_1_-expressing striatonigral MSNs in mice and again showed reduced locomotor activity and motor learning deficits on a rotarod task. Drosophila models of dysregulated dDA1 (Drosophila D_1_ equivalent) function also manifest motor and learning abnormalities, but again there is an inconsistent pattern among different models. Two strains generated either by insertion of transposable elements or gene inversion showed impaired responses to conditioning stimuli, rescued by reinstating dDA1 expression [37]. In contrast, reduction of dDA1 expression by RNA interference resulted in increased larval locomotion, [38] whereas a similar approach significantly reduced motor activity in adult flies [39]. Despite these phenotypic differences, mouse and Drosophila models collectively appear to support an essential role for *DRD1* in motor development.

We have shown that the response of the *DRD1*-T37K variant to agonists is greatly reduced. While our data strongly support a role for impaired ligand binding, we cannot exclude an additional independent effect of the variant on GPCR activation, given the near-total lack of response to agonists at most concentrations. Dopamine binding to D_1_ triggers conformational changes to allow GTP docking, dissociation of the G-protein α-subunit and adenylyl cyclase activation [30]. Recruitment of various scaffolding proteins, such as arrestins, is also required to successfully initiate the signalling cascade [32]. It is plausible that the confirmational change observed due to the T37K variant not only disrupts ligand binding, but also leads to an altered structure for G-protein binding and/or scaffold recruitment, resulting in defective signal transduction.

Given the pharmacoresistant nature of our proband’s disease, including non-responsiveness to levodopa, we explored whether different dopaminergic agonists might increase mutant protein function. Unfortunately, we observed no improvement with any that we tried. It is likely that the defects in ligand binding and G-protein signal transduction cannot be mitigated by orthosteric dopaminergic agonists: positive allosteric modulators, currently in preclinical studies, might possibly be beneficial [40,41]. It would also be useful to examine the effect of agonists in a more sophisticated model system—for example, neurons developed from patient-derived induced pluripotent stem cells (iPSC), striatal-cortical organoid assembloids, or a mouse model—where the full complexity of the dopaminergic network, including other receptor subtypes, would be present.

It is possible that this single-gene disorder could be a candidate for gene-replacement therapy in the future, a strategy already employed for other disorders of monoamine neurotransmission [42,43,44] Gene replacement therapy for AADC deficiency is already in the clinic, and involves stereotactic injection of an AAV2 viral vector containing human *DDC* either into the putamen or midbrain [42,43]. In theory, DRD1 deficiency might also be amenable to a similar targeted gene transfer approach. Although expression of DRD1 is less anatomically concentrated than that of *DDC,* it is strongly enriched in the basal ganglia, especially the striatum (with broad connections to striatum, cortex, nucleus accumbens and putamen), which would therefore probably be the preferred brain target.

In summary, we describe an autosomal recessive disease associated with loss-of-function variants in *DRD1*. Our work highlights the crucial importance of D_1_ in normal motor control. Future work should include identification of additional patients to confirm that association, and study of this and other variants in more complex animal or neuronal models. In addition, in the future, a knock-in or knock-out mouse model of DRD1-related disease would be useful to further understand the underlying disease mechanisms.

## Figures and Tables

**Figure 1 cells-12-01046-f001:**
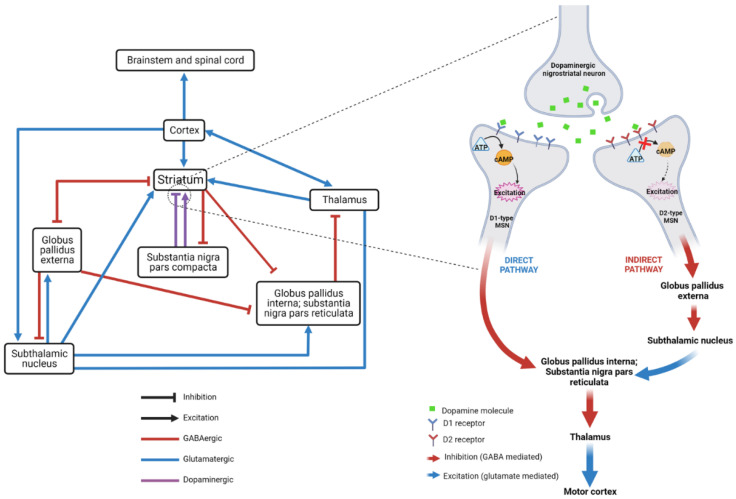
Schematic representation of direct and indirect pathways of the basal ganglia. Left: The direct and indirect pathways regulate voluntary movement and form a key part of the cortico-basal ganglia-thalamic network. The striatum receives inputs including dopaminergic fibres from the substantia nigra, as well as GABAergic (inhibitory) inputs from the globus pallidus pars externa and glutamatergic (excitatory) inputs from the cortex, thalamus and subthalamic nucleus. In turn, it provides inhibitory output via GABergic medium spiny neurons (MSNs) to the globus pallidus pars interna and externa and the substantia nigra. The balance of these outputs determines the relative activity of the direct (prokinetic) and indirect (antikinetic) pathways by modulating thalamic excitatory output to the motor cortex. Right: Dopamine in the striatum has opposite effects depending on whether the postsynaptic MSN expresses D_1_ or D_2_ receptors, exciting the former—which contribute to the direct pathway—and inhibiting the latter—which contribute to the indirect pathway.

**Figure 2 cells-12-01046-f002:**
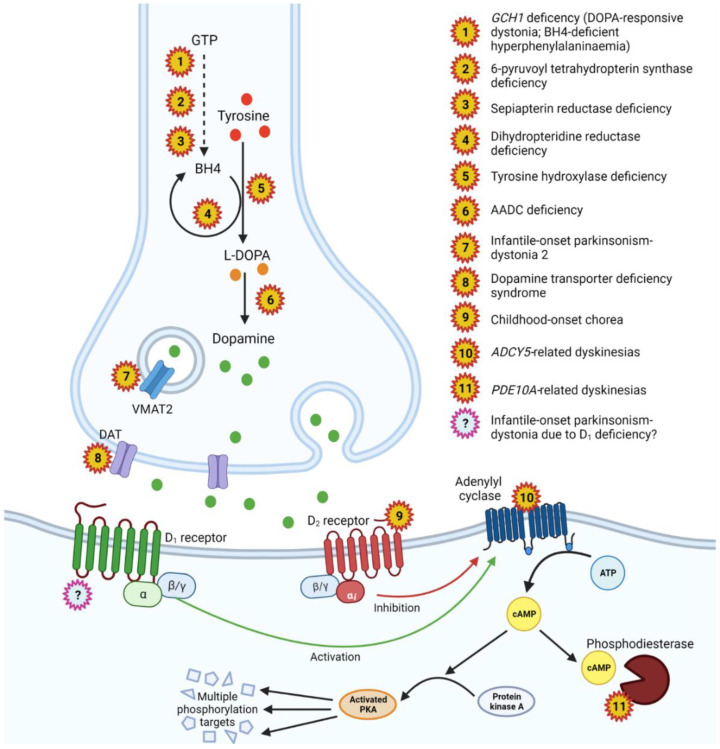
Dopamine pathway in the pre-synaptic dopaminergic neuron: key enzymes, proteins and associated diseases. Schematic representation of a dopaminergic synapse. Dopamine is synthesised from tyrosine in the presynaptic cell: tyrosine is converted to L-dopa by tyrosine hydroxylase, with tetrahydrobiopterin as a cofactor, then L-dopa is decarboxylated to dopamine by aromatic l-amino acid decarboxylase (AADC). Dopamine is packaged into presynaptic vesicles by the vesicular monoamine transporter 2 (VMAT2), before being released into the synapse. On the post-synaptic membrane, dopamine binds D_1_-like and D_2_-like receptors, causing the activation or inhibition of adenylyl cyclase, leading to increase or decrease of intracellular cAMP and downstream signalling pathways mediated by protein kinase A. Stars indicate genetic diseases associated with defects in these pathways.

**Figure 3 cells-12-01046-f003:**
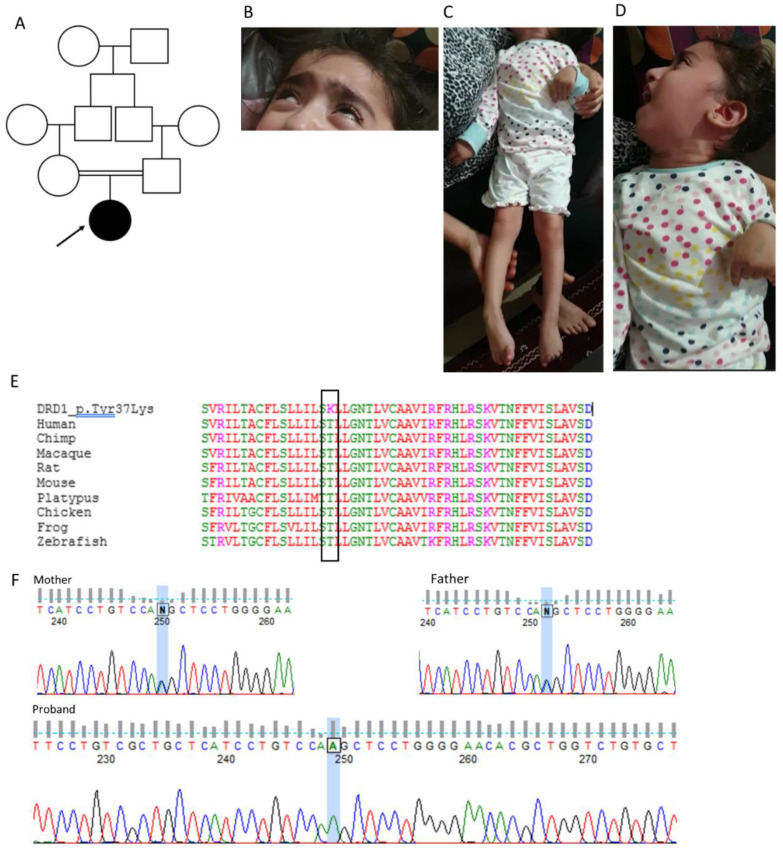
Clinical features and genetic findings in proband. (**A**) Family pedigree, with affected individuals indicated by black shading. (**B**–**D**) Images of proband aged approximately four years depicting phenotype. (**B**) Bilateral fixed upwards eye deviation during an oculogyric crisis. (**C**) Dystonic posturing of trunk and limbs; note striatal toe on the right. (**D**) Opisthotonic posturing and dystonic jaw opening. (**E**) Alignment of cross-species *DRD1* protein sequences in human, chimp, macaque, rat, mouse, platypus, chicken, frog and zebrafish showing conservation of the amino acid and surrounding region. Black box indicates relevant amino acid. (**F**) Sanger sequencing analysis confirming the *DRD1* variant. Sequence chromatograms show heterozygous variants at position c.110 in both parents, and a homozygous variant in proband.

**Figure 4 cells-12-01046-f004:**
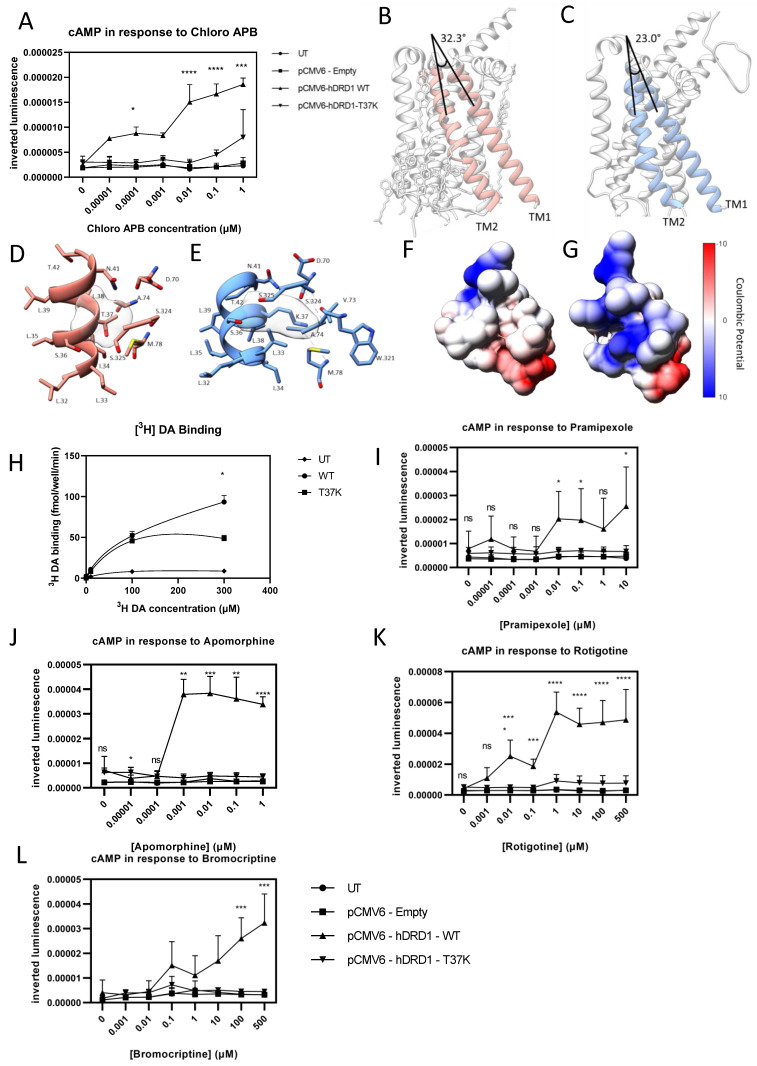
Effect of T37K variant on cAMP production and ligand binding. (**A**) HEK-293T cells transfected with either *DRD1*-WT or *DRD1*-T37K were treated with increasing concentrations of Chloro APB for 10 min and cAMP levels recorded. Graph displays the mean inverted luminescence +/− SEM (n = 6). ns = not significant, * *p* < 0.05, ** *p* < 0.01, *** *p* < 0.005, **** *p* < 0.001 (2-way ANOVA between *DRD1*-WT and *DRD1*-T37K). (**B**–**G**) Structures of DRD1 wildtype and modelled T37K mutant proteins. The side view of the DRD1 proteins showing the angles between the helices TM1 and TM2 with the wildtype protein (PDB ID: 7JVP) in pink (**B**) and T37K mutant in blue (**C**). The orientation of T37 (**D**) and K37 (**E**) in the DRD1 wildtype (pink) and T37K mutant proteins (blue). The amino acids are represented in single letter codes and coloured by atom-type. Coulombic potential-based surface rendering of the wildtype (**F**) and the T37K mutant (**G**) proteins around the sites depicted in (**C**,**D**). Red, blue and white correspond to negative, positive and no charge on the residues, respectively. The graphical representation of the protein structures was created using the molecular graphics program Chimera (http://www.cgl.ucsf.edu/chimera/ accessed on 1 December 2022). (**H**) DA binding to either D_1_-WT or D_1_-T37K was investigated using increasing concentrations of (^3^H) DA. Data are presented as the mean +/− SEM relative to untransfected treated cells (2-way ANOVA, n = 6, *p* < 0.05). (**I**–**L**) Transfected HEK-293T cells treated with clinically relevant D_1_ agonists (**I**) pramipexole, (**J**) apomorphine, (**K**) rotigotine and (**L**) bromocriptine. Graph displays the mean inverted luminescence +/- SEM (n = 6). ns = not significant, * *p* < 0.05, ** *p* < 0.01, *** *p* < 0.005, **** *p* < 0.001 (2-way ANOVA between D_1_-WT and D_1_-T37K).

## Data Availability

The authors confirm that the data supporting the findings of this study are available within the article and/or its Appendix A.

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
