# Peer review of "Loss-of-Function Variants in DRD1 in Infantile Parkinsonism-Dystonia"

_cells, 2023, doi:10.3390/cells12071046_

Round 1
Reviewer 1 Report
Summary
This article describes a new mutation responsible for an inherited childhood dystonia. Genetic and functional analysis reveals the DR1 mutation [c.110C>A, (p.T37K)] in homozygosity as responsible for the disease. The identification of the candidate gene and the specific mutation will allow the molecular diagnosis of new cases and also the detection of carriers in the population.
General concept comments
The manuscript is clear and concise, and all the figures clarify are easy to interpret and understand the text. Specifically schemes (Figure 1 and 2) are very well done and facilitate the understanding of basal ganglia circuit and the dopamine relevance.
The hypothesis is well constructed and the choice of methods to confirm it is appropriate. The data are interpreted appropriately and consistently throughout the manuscript. After molecular genetic analysis, the choice of Dr1 (p.T37K) as the main candidate responsible for the disease is well argued. The functional analysis is well performed and confirms the possibility that the mutation [c.110C>A, (p.T37K)] plays a relevant role in the function of the protein.
The methods are well described and sufficiently detailed to allow reproducibility. The references cited are relevant, most are recent publications and do not include an excessive number of self-citations.
The mutation DR1 [c.110C>A, (p.T37K)] is present in homozygous in the proband and in heterozygous in both parents. Structure-function modeling studies predicted the altered configuration of the mutated DR1 protein, which was confirmed by functional studies. The conclusion reached by the authors is consistent and well argued. The mutated receptor can´t bind dopamine, the proband has symptoms related to the dopaminergic system, which allows to conclude that this DR1 mutation is the responsible for dystonia in the patient.
The main weakness is that article analyzed a single case. Study of other patients with the DR1 mutation [c.110C>A, (p.T37K)] will confirm that it is responsible for this hereditary dystonia. However, it is important to publish this first case, as giving it visibility will help to find other similar cases.
As finding other patients with the same mutation is difficult, one possibility to confirm the results is to replicate the specific mutation [c.110C>A, (p.T37K)] in transgenic mice and then analyze the phenotype.
Reviewer 2 Report
1. The use of HEK-293T cells should be illustrated and explained more in the discussion or method part.
2. Had the mutation of DRD1 been verified in the other family members? (except parents)
Reviewer 3 Report
The authors report a patient with classical infantile parkinsonism-dystonia in whom they identified a homozygous variant in DRD1 resulting in loss of function of the D1 receptor.
They describe only one patient so the authors acknowledge that the gene/disease relationship can not be fully confirmed.
The authors suspected a monoamine neurotransmitter disorder. However, CSF monoamine neurotransmitter levels and AADC enzyme activity were normal and neurometabolic investigations, a genetic panel for childhood-onset movement disorders, EEG and imaging including MRI and DaTscan were non-contributory.
They identified the DRD1 recessive homozygous variant (T37K) via WGS and panel-free analysis among 19 homozygous variants detected in the proband.
They performed some in vitro studies thus showing that i) DRD1-T37K shows no statistical differences in total protein expression and cell surface localisation, ii) DRD1-T37K affects the cAMP response on D1 activation and iii) DRD1-T37K function cannot be rescued by dopamine agonists.
They also confirmed in vitro the altered ligand binding prediction obtained through homology modelling of DRD1-T37K.
They also mention the DRD1 KO studies made in mice and drosophila and acknowledge that the results if these studies are in support for DRD1 role in motor development.
Overall, this is a very nice paper. The strategy is straightforward and the results are clearly presented. In conclusion this work adds support for the role of DRD1 in motor control.
